# Systematic Review and Meta-Analysis of Surgical Treatment for Isolated Local Recurrence of Pancreatic Cancer

**DOI:** 10.3390/cancers13061277

**Published:** 2021-03-13

**Authors:** Simone Serafini, Cosimo Sperti, Alberto Friziero, Alessandra Rosalba Brazzale, Alessia Buratin, Alberto Ponzoni, Lucia Moletta

**Affiliations:** 1Department of Surgery, Oncology and Gastroenterology, 3rd Surgical Clinic, University of Padua, Via Giustiniani 2, 35128 Padua, Italy; simone.serafini@ymail.com (S.S.); alberto.friziero@aopd.veneto.it (A.F.); lucia.moletta@unipd.it (L.M.); 2Department of Statistical Sciences, University of Padua, Via Cesare Battisti 241, 35121 Padua, Italy; brazzale@stat.unipd.it; 3Department of Biology, University of Padua, Viale G. Colombo 3, 35131 Padua, Italy; buratin.alessia1@gmail.com; 4Department of Radiology, Padua General Hospital, Via Giustiniani 2, 35128 Padua, Italy; alberto.ponzoni@aopd.veneto.it

**Keywords:** isolated local recurrence, pancreatectomy, pancreatic cancer, pancreatic remnant, recurrence, redo surgery

## Abstract

**Simple Summary:**

Recurrences after primary resection of pancreatic cancer are generally treated with chemotherapy or best supportive care. Despite some reports of encouraging results after the re-resection of recurrences, the real role of surgery in this setting remains unclear. The aim of our systematic review and meta-analysis was to define the benefit of surgery in the case of isolated local recurrence. The data collected on 431 patients suggest an overall survival benefit of 29 months for patients re-operated compared to patients given medical therapies. In selected patients with recurrent pancreatic cancer, resection is safe and feasible, and may offer a survival advantage.

**Abstract:**

Purpose: To perform a systematic review and meta-analysis on the outcome of surgical treatment for isolated local recurrence of pancreatic cancer. Methods: A systematic review and meta-analysis based on Preferred Reporting Items for Systematic Reviews and Meta-analyses (PRISMA) guidelines was conducted in PubMed, Scopus, and Web of Science. Results: Six studies concerning 431 patients with recurrent pancreatic cancer met the inclusion criteria and were included in the analysis: 176 underwent redo surgery, and 255 received non-surgical treatments. Overall survival and post-recurrence survival were significantly longer in the re-resected group (ratio of means (ROM) 1.99; 95% confidence interval (CI), 1.54–2.56, *I*^2^ = 75.89%, *p* = 0.006, and ROM = 2.05; 95% CI, 1.48–2.83, *I*^2^ = 76.39%, *p* = 0.002, respectively) with a median overall survival benefit of 28.7 months (mean difference (MD) 28.7; 95% CI, 10.3–47.0, *I*^2^ = 89.27%, *p* < 0.001) and median survival benefit of 15.2 months after re-resection (MD 15.2; 95% CI, 8.6–21.8, *I*^2^ = 58.22%, *p* = 0.048). Conclusion: Resection of isolated pancreatic cancer recurrences is safe and feasible and may offer a survival benefit. Selection of patients and assessment of time and site of recurrence are mandatory.

## 1. Introduction

Pancreatic ductal adenocarcinoma (PDAC) was the fourth cause of cancer-related death in the United States and in Italy in 2018, with an estimated 55.440 and 13.300 new cases diagnosed, and 44.300 and 11.463 related deaths, respectively [1,2]. Surgical resection continues to be the only chance of cure, but the 5-year survival rate remains low, ranging from 19% to 27% [3,4]. Such disappointing results are justified by the aggressive biology of pancreatic cancer and the high rate of recurrence (up to 80%) even after radical resections [5]. Recurrent pancreatic cancer poses a challenge for clinicians and is commonly treated with chemotherapy or best supportive care. Unlike other cancers, re-surgery for relapsing pancreatic cancer is not generally considered an option because evidence regarding its benefits is lacking, and isolated tumor recurrences amenable to resection are relatively uncommon. An isolated local recurrence (ILR) is usually defined as a tumoral recurrence localized to the posterior resection margin, the pancreatic remnant, or the locoregional lymph nodes. Some authors have recently reported encouraging results of surgical management of pancreatic cancer recurrence in selected patients [6,7,8,9].

The aim of the present study was to conduct a systematic review and meta-analysis of the outcome of redo-surgery for patients with isolated local recurrent PDAC after initial pancreatectomy.

## 2. Materials and Methods

### 2.1. Study Selection

A systematic literature search was conducted using PubMed, Scopus, and Web of Science to identify all studies published up to 30 November 2020 regarding the recurrence of pancreatic cancer after surgery. The search terms used were “pancreatic cancer/neoplasm/adenocarcinoma”, “recurrence”, “surgery/pancreatectomy/redo surgery/completion pancreatectomy”. The articles found were used to broaden the search, and all emerging abstracts, studies, and citations were reviewed. The reference lists of all the studies considered were also screened for any other potentially relevant papers.

### 2.2. Inclusion and Exclusion Criteria

The following inclusion criteria were considered for the studies: (1) they reported on patients with histologically proven ILR PDAC treated surgically with curative intent, with or without (neo) adjuvant chemotherapy and/or radiotherapy; (2) they provided data on patients reoperated for recurrent PDAC after initial pancreatectomy, and their long-term outcomes; (3) they were written in English. The following exclusion criteria were considered: (1) reviews without original data or animal studies; (2) absence of individual patient data; (3) duplications; (4) lack of long-term data; and (5) in the event of successive publications by the same group, only the most detailed study was included.

Three independent reviewers (SS, ARB, AB) extracted the data using standardized data forms. All data from each eligible study were entered in a dedicated spreadsheet (Excel 2007, Microsoft Corporation^®^, Padua, Italy). Disagreements between the reviewers were solved by discussion and consensus. The following data were collected: title, first author, year of publication, characteristics of study population, study design, number of patients who underwent re-resection, disease-free interval (DFI), overall survival (OS), and post-recurrence survival (PRS). The articles included in this review were chosen in accordance with the Preferred Reporting Items for Systematic Reviews and Meta-Analyses (PRISMA) guidelines [10].

### 2.3. Terminology and Definitions

Disease-free interval (DFI) was defined as the interval between the date of primary tumor resection and the date of recurrence. Overall survival (OS) was defined as the time between the primary tumor resection and death or latest follow-up. Post-recurrence survival (PRS) was the time interval between recurrence detection or reoperation and death or latest follow-up.

### 2.4. Statistical Analysis

The statistical methods of this study were reviewed by two authors (BAR and BA). Two meta-analyses were conducted in line with the Cochrane Collaboration guidelines on the Meta-analysis of Observational Studies in Epidemiology [11,12]. The first analysis focused on OS in months; the second focused on survival in months after the recurrence of PDAC. The data used for the meta-analyses are summarized in Table 1. Survival was retrieved from the published studies as median values and ranges. Where not stated explicitly, these values were calculated from the data reported in the papers or extrapolated from the Kaplan-Meier (K-M) plots. Since all articles used in this analysis reported only the size of the study groups (without standard errors), all median survival times were converted into means and variances using a dedicated statistical algorithm [13].

The survival data were pooled for the analysis of either the mean difference (MD) or the logarithm of the ratio of means (ROM) [14]. Values of MD > 0 or ROM > 1 indicate a higher survival rate for patients who underwent re-resection. Cochran’s Q statistic and the *I*^2^ statistic were used to test between-study heterogeneity [15]. If the Q statistic was significant at the 0.1 level, the summary effect and corresponding 95% confidence interval (CI) were obtained with the Mantel-Haenszel random effects model [16]. For *I*^2^ < 50%, between-study heterogeneity was judged to be low moderate; for *I*^2^ ≥ 50%, it was considered substantial. The point estimate of MD and ROM was considered statistically significant when *p* was < 0.05. A cumulative meta-analysis was also run to test the stability of the pooled endpoint estimates. Publication bias was assessed visually using a funnel plot, and the number of missing studies was estimated using the trim-and-fill method [17,18]. All analyses were conducted using R version 3.5.2 [19].

**Table 1 cancers-13-01277-t001:** Studies included in the meta-analysis. OS and PFR are compared for each study (significant difference for *p* value < 0.05). Pts: number of patients; OS: overall survival; PRS: post recurrence survival; * extrapolated from K-M plot; ^ extrapolated from confidence interval; NR: not reported.

Author	Treatment	Pts	OS (Range)	*p* Value	PRS (Range)	*p* Value
Strobel et al. [20]	RESECTED *	41	NR	NR	26 (1–60)	<0.01
	UNRESECTED *	16	NR		10.8 (1–36)	
Miyazaki et al. [21]	RESECTED *	11	78.2 (17–107)	<0.001	25 (3–61)	<0.01
	UNRESECTED *	159	20.3 (5–103)		9.3 (3–75)	
Hashimoto et al. [22]	RESECTED	8	72 (36–129)	NR	17 (10–85)	NR
	UNRESECTED	2	30 (28–32)		10 (9–11)	
Nakayama et al. [23]	RESECTED ^	11	70 (19–70)	=0.02	44 (11–44)	=0.01
	UNRESECTED ^	35	25 (15–35)		11 (6–25)	
Yamada et al. [24]	RESECTED *	90	26 (4–60)	=0.012	NR	NR
	UNRESECTED *	24	14 (2–60)		NR	
Kim et al. [25]	RESECTED *	15	NR	NR	28 (5–57)	=0.01
	UNRESECTED *	19	NR		12 (4.5–35)	

## 3. Results

The study search and selection strategy are shown in the flow chart in Figure 1. The preliminary literature search identified 1326 studies matching the initial search criteria. After screening, six studies were ultimately included in the quantitative synthesis (meta-analysis) [20,21,22,23,24,25].

It was not possible to include DFI in this analysis because this information was lacking in most studies. Likewise, due to the lack and fragmentation of precise data about the time to recurrence, it was not possible to estimate and compare conditional survival of resected and non-resected patients. Four reports [21,22,23,24] were included for OS and five for PRS [20,21,22,23,25] analysis. All six studies were retrospective and concerned a total of 431 patients with recurrent pancreatic cancer after primary pancreatic resection: 176 treated with re-resection, and 255 given non-surgical removal of recurrence. The characteristics of the selected studies are summarized in Table 1.

After surgery for recurrent PDAC, the mortality rate reported was 1.1% (2/176 patients) [20,25]; the morbidity rate ranged from 6% [25] to 33% [24].

A random-effects meta-analytical model was used for all variables. The random-effects method was always chosen because Cochran’s Q statistic proved statistically significant at the *p* < 0.05 level for all meta-analyses, with a borderline situation for the analysis focusing on mean survival after recurrent PDAC, for which it was *p* = 0.048.

OS was almost twice as long in patients reoperated for ILR PDAC than in patients given non-surgical treatments (ROM 1.99; 95% CI, 1.54–2.56, *I*^2^ = 75.89%, *p* = 0.006). The median survival benefit for patients who underwent re-resection was 28.7 months (MD 28.7; 95% CI, 10.3–47.0, *I*^2^ = 89.27%, *p* < 0.001), as shown in Figure 2.

Median PRS was significantly longer for the group treated surgically than for the patients not re-resected (ROM = 2.05; 95% CI, 1.48–2.83, *I*^2^ = 76.39%, *p* = 0.002): it was 15.2 months longer in the former than in the latter group (MD 15.2; 95% CI, 8.6–21.8, *I*^2^ = 58.22%, *p* = 0.048) (Figure 3).

The cumulative meta-analysis demonstrated that the benefit after pancreatic re-resection settles very quickly for both OS and PRS. Funnel plots provided some evidence of publication bias. One study was estimated to be missing on the left for OS, while no study seems to be missing for PRS (Figure 4 and Figure 5).

## 4. Discussion

Pancreatic ductal adenocarcinoma is recognized as a major cause of cancer-related deaths for early metastasis, extensive invasion, and poor prognosis. At diagnosis, 50% of patients present with synchronous metastases, and further 30% present with locally advanced disease, who are not suitable for upfront surgery [1]. Moreover, despite radical resection, PDAC frequently relapses, and the clinical management of recurrences is troublesome. The well-accepted treatment for recurrent PDAC is still chemotherapy, whenever feasible. Recent studies on the surgical treatment of recurrent PDAC in selected patients reported encouraging results in terms of survival, with negligible surgical morbidity and mortality rates.

Strobel et al. [20] conducted a prospective cohort study with patients with pancreatic cancer recurrence and assessed perioperative outcome, survival, and prognostic parameters. In this series, 57 patients underwent surgery for histologically proven ILR after R0/R1 resection of PDAC. ILR was resected in 41 patients. Most resections were carried out for extrapancreatic recurrences. A pancreatic re-resection was performed in 24 patients (44%). In 19 cases, a total pancreatectomy was necessary; segmental resections were possible in only 5 patients. In 11 (20%) patients with ILR in paracaval or interaortocaval lymph nodes, a simple excision with lymphadenectomy was performed. A total of 36 (63%) patients with ILR had a recurrence in close touch with visceral arteries (SMA or the celiac trunk). In these cases resection was only attempted if the arteries were not directly involved. No data about the number of R1/R0 resection after the first operation were reported. The authors [20] also assessed the potential effect of intraoperative radiation therapy (IORT). A total of 22 patients underwent surgical resection and IORT. In 16 patients, the ILR was considered not resectable because of infiltration of the mesenteric vessels; 10 of these patients received IORT (10–15 Gray). In patients with resection of ILR, the subgroup with IORT demonstrated a shorter survival than patients without IORT (17.0 vs. 29.6 months of median survival). In contrast, patients with unresectable ILR had significantly better survival with IORT (15.1 vs. 4.3 months of median survival). They concluded that benefit of percutaneous and intraoperative radiotherapy warrants further evaluation.

Miyazaki et al. [21] reported on 284 consecutive patients with pancreatic cancer who underwent initial pancreatectomy with curative intent (R0 and R1 resection). A total of 170 patients were diagnosed with recurrent pancreatic cancer, but only 11 (16.4%) developed ILR. Two out of eleven were R1 at the time of the first operation.

Hashimoto et al. [22] retrospectively analysed the survival and pathological findings of 10 patients who developed remnant pancreatic cancer. The authors performed a pyrosequencing assay for KRAS (codon 12) mutations and immunohistochemistry for MUC1/MUC2, and compared the histological diagnosis of the initial tumor and the remnant pancreatic cancer in the resected group. The results indicated that four cases might have developed local recurrence of the primary lesions, and the other four cases might have developed new primary lesions. It is important to point out that only one case out of ten was R1 resection at the time of the first operation.

Nakayama et al. [23] compared the survival outcomes of patients who developed isolated local vs. distant recurrence. In this subset of patients, only 3 out of 46 patients were considered R1 after the first pancreatic resection. Median survival after the recurrence was longer in the patients with ILR than in those with distant metastasis (44 vs. 13 months, *p*-value < 0.05).

Yamada et al. [24] conducted a multicenter survey of patients diagnosed with ILR in the remnant pancreas. Data from 114 patients were collected in a retrospective manner. Although their multivariate analysis could not identify any independent prognostic factors, the univariate analysis showed that excision of ILR, age (<65 years), body mass index (>20 kg/m^2^), tumor dimensions (<20 mm), and distance from the pancreatic resection margin (>10 mm) were statistically significant positive prognostic factors. In this study, there was no correlation between the R1 margin at the time of first operation and distance to the pancreatic stump of ILR, which implies that the onset of a tumor in the remnant pancreas should not always be considered as a consequence of intrapancreatic colonization of the primitive cancer, as supported by Hashimoto et al. [22].

Kim et al. [25] reported on a cohort of 1610 consecutive patients with pancreatic cancer who underwent initial pancreatectomy with curative intent between January 2000 and December 2014 at Asian Medical Centre, Seoul, Korea. A total of 1346 patients were diagnosed with recurrent pancreatic cancer, but only 197 (14.6%) and 34 (2.5%) of these patients had isolated recurrence and ILR, respectively. Moreover, the authors performed a survival analysis according to the recurrence pattern. Survival after recurrence was better in patients who underwent resection of isolated recurrence in the remnant pancreas (median 28 vs. 12 months) and lung (median 36.5 vs. 9.5 months) than in those who did not undergo resection.

In a previously published systematic review by Moletta et al. [26] about the role of surgical resection for recurrent PDAC, an overall survival benefit after resection compared to non-resected patients was reported.

This raises some questions. First, is surgery for recurrent PDAC definitely worthwhile? If so, can it be proposed for all sites of recurrence? Which is the optimal treatment for recurrent PDAC: chemotherapy or surgery?

In this study, we confirmed the potential benefit of surgery, applying a quantitative analysis and statistical significance to the data previously reviewed. We analyzed six studies and compared the results of surgery vs. medical treatment (chemotherapy or best supportive care) for patients with recurrent PDAC. The results of our analysis confirmed that redo surgery for recurrent tumor offers a survival advantage for selected patients with a very low risk of perioperative mortality and an acceptable morbidity rate. The survival benefit (patients re-operated for ILR compared to patients given medical therapies) was estimated as about 29 and 15 months in terms of OS and PRS, respectively.

The resection of recurrent disease seems to be feasible and safe, and should be considered for selected patients with isolated pancreatic cancer local recurrences. Among the various sites affected, surgery for recurrences in the pancreatic remnant seems to be associated with a better outcome. Zhou et al. [8] reviewed the English literature until June 2016, collecting 19 articles on 55 patients who had complete pancreatectomy for relapsing PDAC. The 1, 3, and 5-year OS rates after second pancreatectomy were 82.2%, 49.2%, and 40.6%, respectively.

To date, only some reports and case series that investigate the benefit of surgery in isolated distant recurrence or in oligo metastatic patients have been published. Despite advances in surgical techniques, pancreatic surgery often results in a positive resection margin status (R). In particular, R0 was defined as a distance from the tumor to the closest resection margin of >1 mm, whereas R1 as a distance of ≤1 mm to the resection margin or margin involved without macroscopic involvement. R1 resection and medial/posterior margin due to perineural invasion, regional lymph node metastases, and systemic spreading at the time of surgery resulted in early recurrences. Based on today’s knowledge, it is not possible to define if ILR are de novo tumors or recurrence of the primary ones. Encouraging survival rates have been reported for isolated lung metastases: the median survival time after lung metastasectomy ranged from 18.6 [9] to 47 months [27]. On the other hand, poor outcomes have been reported after the resection of liver or peritoneal metastases [28,29]. New randomized clinical trials should be conducted to better define the role of surgery in this setting.

We cannot clearly answer the question regarding the optimal treatment for recurrent PDAC because of the lack of controlled studies comparing the outcomes of patients who undergo surgery with those given chemotherapy. In the era of multi-agent systemic therapy, survival for pancreatic cancer has globally increased. However, even if many different randomized clinical trials for borderline, locally advanced, and “de novo” metastatic PDAC are published confirming the survival benefit of new regimen, in the case of recurrent PDAC, such studies are missing, and there is currently no scientific evidence supporting a specific treatment. Some authors have shown improvement in survival after treatment of recurrent PDAC with intensified regimes including FOLFIRINOX (FFN) and nab-paclitaxel plus gemcitabine (GEMNAB) compared with single agent chemotherapy. Gbolahan et al. [30] reported that administration of FFN or GEMNAB compared with single agent chemotherapy was associated with a statistically significant survival benefit, with a median OS of 14 (95% CI 9–17) vs. 8 (95% CI 6–12) months. Javed et al. [31], in a recent retrospective multi-center European study, did not report any significant differences in terms of survival between FFN/GEMNAB and any other combination of gemcitabine- or 5-fluorouracil-based regimen: polichemotherapy was always superior when compared with gemcitabine monotherapy with a variable median OS of 7.9–9.9 vs. 4.9 (95% CI 4.4–5.6) months. Kawaida et al. [32] reported an objective response rate of 13.6% and a progression-free survival of 7.2 months after administration of GEMNAB with an important hematological toxicity rate of 72.7% (grade 3–4).

It is reasonable to believe that the combination of new chemotherapy regimens and surgery in fit patients could improve the outcome of recurrent PDAC. Given the heterogeneous chemotherapy regimens used, the small numbers of patients included in different studies, and the fact that most patients present with multiple sites of relapse and frequently in poor health, inevitably, the surgical option can only be offered to a few, very selected patients.

Our study has some limitations to consider. The relatively small number of studies analyzed and their heterogeneity and retrospective nature entail a significant risk of selection bias. The lack of data in some studies also prevented us from measuring disease-free interval and compare conditional survival to obtain a more accurate picture of patients’ outcomes. We therefore were unable to investigate the potential role of a number of factors including patient’s characteristics, baseline tumor burden and stage, neoadjuvant chemotherapy, morbidity from surgical resection and adjuvant therapy, comorbidity, and functional status on failure to receive therapy. All these aspects can lead to a selection bias that it is difficult to avoid, according to the rarity of ILR in PDAC. These important questions should be further addressed in new prospective and multicentric studies.

## 5. Conclusions

To our knowledge, this is the first meta-analysis comparing the outcome of patients with ILR PDAC following resection or sequential chemotherapies. In selected patients with recurrent pancreatic cancer, resection is safe and feasible, and may offer a survival advantage. Surgery should be considered as part of the multimodality management of relapsing pancreatic cancer. An accurate patient selection, considering the site and time of recurrence, and a multidisciplinary approach are essential to choose the best appropriate treatment.

## Figures and Tables

**Figure 1 cancers-13-01277-f001:**
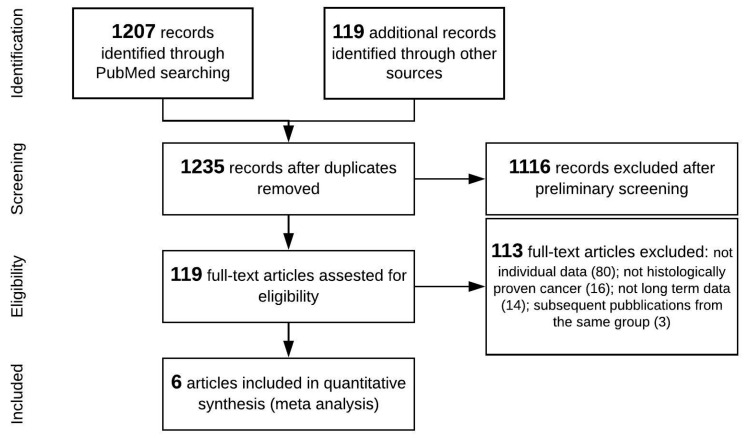
Preferred Reporting Items for Systematic Reviews and Meta-Analyses (PRISMA) flow diagram.

**Figure 2 cancers-13-01277-f002:**
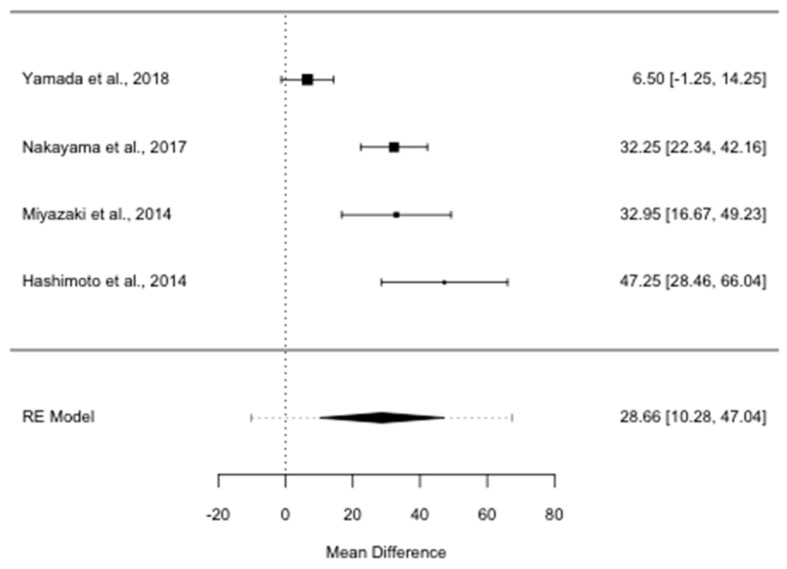
Forest plot for overall survival mean difference in months. RE: random-effects.

**Figure 3 cancers-13-01277-f003:**
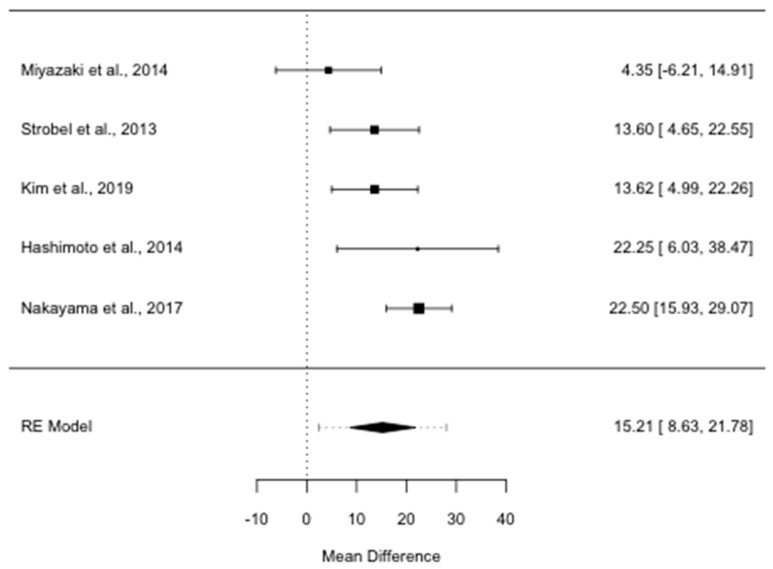
Forest plot for post recurrence survival mean difference in months. RE: random-effects.

**Figure 4 cancers-13-01277-f004:**
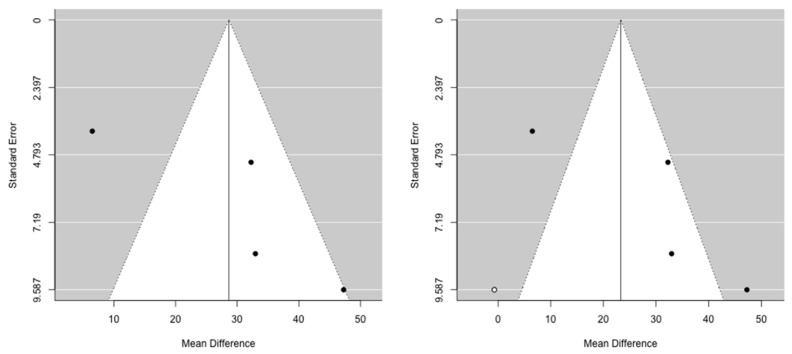
Funnel plots for mean difference in overall survival. Left: original data; Right: imputation of missing studies.

**Figure 5 cancers-13-01277-f005:**
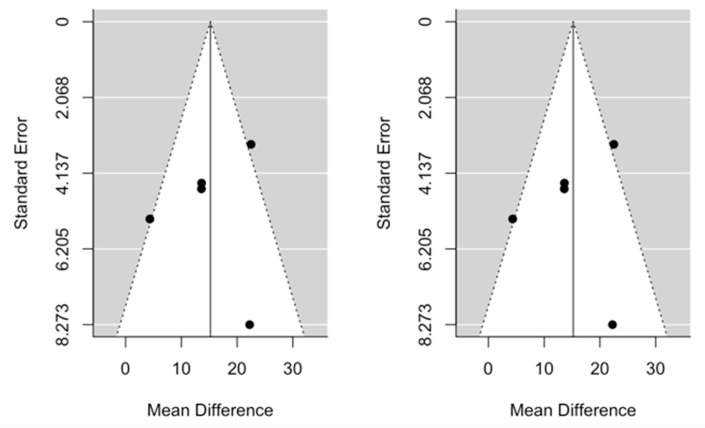
Funnel plots for mean difference in post recurrence survival. Left: original data; Right: imputation of missing studies.

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
