# Peer review of "Systematic Review and Meta-Analysis of Surgical Treatment for Isolated Local Recurrence of Pancreatic Cancer"

_cancers, 2021, doi:10.3390/cancers13061277_

Round 1

Reviewer 1 Report

In the manuscript "Systematic Review and Meta-Analysis of Surgical Treatment for Isolated Local Recurrence of Pancreatic Cancer" by Simone Serafini et al., the authors revise and conduct a meta-analysis on the benefit of a new surgery in cases of recurrence of pancreatic cancer, which is known to have an aggressive course, with global dismal prognosis.

The paper is well-written, the ideas clearly exposed and the meta-analysis correctly performed. Nevertheless, there are some minor aspects that merit the attention of the authors, which are the following:

  1. The initial introductory sentence in the discussion section should be improved to more accurately portray the aggressiveness and dismal prognosis of pancreatic cancer.

  1. In line 238 there is a typo in “published”.

Therefore, I consider that this study is very interesting and extremely relevant for publication in Cancers after the correction of the aspects highlighted above.

Author Response

Dear Editor, Dear Reviewer I,

We would like to thank the reviewer for the constructive comment to our manuscript “Systematic Review and Meta-Analysis of Surgical Treatment for Isolated Local Recurrence of Pancreatic Cancer" (ID: Cancers-1126780).

We herewith provide a point-by-point response to the reviewer’s comments and submit the revised manuscript for possible publication in Cancers Special Issue “Surgical Treatment of Pancreatic Ductal Adenocarcinoma”.

  • 1 Reviewer’s comment: The initial introductory sentence in the discussion section should be improved to more accurately portray the aggressiveness and dismal prognosis of pancreatic cancer.

Response: As the reviewer suggested, we added in the discussion the sentences "Pancreatic ductal adenocarcinoma is recognized as a major cause of cancer-related deaths for early metastasis, extensive invasion and poor prognosis. At diagnosis, 50% of the patients present with synchronous metastases and further 30% present with locally advanced disease, which are not suitable for upfront surgery. Moreover, despite radical resection, PDAC frequently relapses and the clinical management of recurrences is troublesome."

  • 2 Reviewer’s comment: In line 238 there is a typo in “published”.

Response: We have corrected the typo.

We have addressed each of the reviewers’ suggestion and highlighted the revisions using the "Track Changes" function in Microsoft Word.

Thank you very much again for your time and kind work on our paper.

Kind Regards,

Prof. Cosimo Sperti

Department of Surgery, Oncology and Gastroenterology,

3rd Surgical Clinic, University of Padua,

Via Giustiniani 2, 35128 Padua, Italy

Phone +390498218845

Fax +390498218821

Reviewer 2 Report

The authors present a current meta-analysis of resection versus non-resection in local recurrence of pancreatic cancer. This is a quantitative follow-up study of a previously published review by the same authors in the same journal.

Introduction and methods are adequate, the PRISMA guideline is followed.

The (only) main finding is that survival in resected patients is much longer than in non-resected. Further subgroup or sensitivity analysis was not feasible due to lack of data. The limitations are clearly stated.

In summary, this is a metaanalysis of an emerging topic in pancreatic cancer treatment, demonstrating a clinically very relevant benefit from re-resection for local recurrence.

I recommend publication after assessment of the following minor issues:

1. the main caveat for this comparison of patients is selection bias. please elaborate this by some more words than just the single mention.

2. is it possible to compare the resected and non-resected patients for the time to recurrence ? this may unravel or disprove one important aspect of selection bias (ie conditional survival).

Author Response

Dear Editor, Dear Reviewer I,

We would like to thank the reviewer for the constructive comment to our manuscript “Systematic Review and Meta-Analysis of Surgical Treatment for Isolated Local Recurrence of Pancreatic Cancer" (ID: Cancers-1126780).

We herewith provide a point-by-point response to the reviewer’s comments and submit the revised manuscript for possible publication in Cancers Special Issue “Surgical Treatment of Pancreatic Ductal Adenocarcinoma”.

  • 1 Reviewer’s comment: The main caveat for this comparison of patients is selection bias. please elaborate this by some more words than just the single mention.

Response: As the reviewer suggested, we added in the discussion the sentences "The lack of data in some studies also prevented us from measuring disease-free interval and compare conditional survival to obtain a more accurate picture of patients’ outcomes. We therefore were unable to investigate the potential role of a number of factors including patient’s characteristics, baseline tumor burden and stage, neoadjuvant chemotherapy, morbidity from surgical resection and adjuvant therapy, comorbidity, functional status on failure to receive therapy. All these aspects can lead to a selection bias that it is difficult to avoid, according to the rarity of ILR in PDAC."

  • 2 Reviewer’s comment: Is it possible to compare the resected and non-resected patients for the time to recurrence ? this may unravel or disprove one important aspect of selection bias (ie conditional survival).

Response: As the reviewer suggested, we added in the discussion the sentences "Likewise, due to the lack and fragmentation of precise data about the time to recurrence, is not possible to estimate and compare conditional survival of resected and non-resected patients."

We have addressed each of the reviewers’ suggestions and highlighted the revisions using the "Track Changes" function in Microsoft Word.

Thank you very much again for your time and kind work on our paper.

Kind Regards,

Prof. Cosimo Sperti

Department of Surgery, Oncology and Gastroenterology,

3rd Surgical Clinic, University of Padua,

Via Giustiniani 2, 35128 Padua, Italy

Phone +390498218845

Fax +390498218821
